# Pharmacological Inhibition of Wee1 Kinase Selectively Modulates the Voltage-Gated Na^+^ Channel 1.2 Macromolecular Complex

**DOI:** 10.3390/cells10113103

**Published:** 2021-11-10

**Authors:** Nolan M. Dvorak, Cynthia M. Tapia, Timothy J. Baumgartner, Jully Singh, Fernanda Laezza, Aditya K. Singh

**Affiliations:** Department of Pharmacology and Toxicology, University of Texas Medical Branch, Galveston, TX 75901, USA; nmdvorak@utmb.edu (N.M.D.); cmtapia@utmb.edu (C.M.T.); tjbaumga@utmb.edu (T.J.B.); jusingh@utmb.edu (J.S.); felaezza@utmb.edu (F.L.)

**Keywords:** voltage-gated Na^+^ (Na_v_) channels, fibroblast growth factor 14 (FGF14), Wee1 kinase, patch-clamp electrophysiology

## Abstract

Voltage-gated Na^+^ (Na_v_) channels are a primary molecular determinant of the action potential (AP). Despite the canonical role of the pore-forming α subunit in conferring this function, protein–protein interactions (PPI) between the Na_v_ channel α subunit and its auxiliary proteins are necessary to reconstitute the full physiological activity of the channel and to fine-tune neuronal excitability. In the brain, the Na_v_ channel isoforms 1.2 (Na_v_1.2) and 1.6 (Na_v_1.6) are enriched, and their activities are differentially regulated by the Na_v_ channel auxiliary protein fibroblast growth factor 14 (FGF14). Despite the known regulation of neuronal Na_v_ channel activity by FGF14, less is known about cellular signaling molecules that might modulate these regulatory effects of FGF14. To that end, and building upon our previous investigations suggesting that neuronal Na_v_ channel activity is regulated by a kinase network involving GSK3, AKT, and Wee1, we interrogate in our current investigation how pharmacological inhibition of Wee1 kinase, a serine/threonine and tyrosine kinase that is a crucial component of the G2-M cell cycle checkpoint, affects the Na_v_1.2 and Na_v_1.6 channel macromolecular complexes. Our results show that the highly selective inhibitor of Wee1 kinase, called Wee1 inhibitor II, modulates FGF14:Na_v_1.2 complex assembly, but does not significantly affect FGF14:Na_v_1.6 complex assembly. These results are functionally recapitulated, as Wee1 inhibitor II entirely alters FGF14-mediated regulation of the Na_v_1.2 channel, but displays no effects on the Na_v_1.6 channel. At the molecular level, these effects of Wee1 inhibitor II on FGF14:Na_v_1.2 complex assembly and FGF14-mediated regulation of Na_v_1.2-mediated Na+ currents are shown to be dependent upon the presence of Y158 of FGF14, a residue known to be a prominent site for phosphorylation-mediated regulation of the protein. Overall, our data suggest that pharmacological inhibition of Wee1 confers selective modulatory effects on Na_v_1.2 channel activity, which has important implications for unraveling cellular signaling pathways that fine-tune neuronal excitability.

## 1. Introduction

The activity of voltage-gated Na^+^ (Na_v_) channels is the primary determinant of the initiation and propagation of action potentials (AP) in excitable cells [1]. Structurally, the pore-forming α subunit of Na_v_ channels, of which nine different isoforms have been described (Na_v_1.1–Na_v_1.9) [1], is comprised of four transmembrane domains (DI-DIV), and each transmembrane domain is comprised of six segments (S1–S6) [2,3]. Despite the central role of this pore-forming α subunit in conferring Na_v_ channel activity, the full physiological function of the Na_v_ channel is dependent upon protein–protein interactions (PPIs) between the Na_v_ channel α subunit and its auxiliary proteins [4,5]. Among these auxiliary proteins, intracellular fibroblast growth factors (iFGF; FGF11–FGF14) represent an important family of accessory proteins that regulate the kinetics and trafficking of Na_v_ channels through direct PPIs with the C-terminal domains (CTD) of different Na_v_ channel isoforms [6,7,8,9,10].

In the central nervous system (CNS), FGF14 is a particularly prominent Na_v_ channel auxiliary protein that regulates resurgent sodium current (*I*_NaR_) [11,12] and differentially modulates transient sodium currents (*I*_Na_) mediated by Na_v_1.2 versus Na_v_1.6 channels [7]. Despite these regulatory effects of FGF14 on different types of neuronal Na+ currents and differential regulation of CNS Na_v_ channel isoforms, less is known about cellular signaling molecules that might modulate FGF14′s effects on these biophysical processes. 

To that end, we previously performed a high throughput screening of kinase inhibitors against different Na_v_ channel macromolecular complexes using an optimized in-cell assay to identify modulators of FGF14′s complex assembly with different Na_v_ channel isoforms [13,14,15]. Through these investigations, in tandem with orthogonal and functional validation modules, we have identified GSK3 [16], AKT [17], and JAK2 [14] as important kinases that regulate FGF14:Na_v_1.6 complex assembly and Na_v_1.6 channel activity; although, the regulatory effects of these kinases on other iFGF:Na_v_ complexes, such as FGF14:Na_v_1.2, are less well characterized. 

In addition to the established roles of the aforementioned kinases in regulating FGF14′s PPI with Na_v_ channel isoforms, our previous investigations have suggested a potential role of Wee1 kinase in regulating PPIs involved in Na_v_ channel macromolecular complex assembly [13,14,17]. Wee1 is a serine/threonine and tyrosine kinase with an established role in regulating the G2-M cell-cycle checkpoint in which the kinase negatively regulates entry of cells into mitosis to allow for DNA repair [18]; although, its effect on neuronal activity is less well-characterized. Providing some insights into the latter, it has been shown that pharmacological inhibition of Wee1 kinase through the employment of some Wee1 kinase inhibitors modulates the complex assembly of various iFGF:Na_v_ channel pairs [13,14]. Additionally, Wee1 kinase activity has been shown to be modulated by GSK3 [19,20,21] and to potentially increase AKT activity [22], which could have important implications for conferring indirect effects on Na_v_ channel activity given the known regulatory effects of these kinases on Na_v_ channel kinetics and trafficking [13,14,16,17]. 

In the present investigation, we focus squarely on assessing this potential regulation of the Na_v_ channel conferred by Wee1 kinase. In particular, we focused on investigating the regulatory effects of Wee1 kinase on the FGF14:Na_v_1.2 and FGF14:Na_v_1.6 complexes on account of FGF14 previously having been shown to confer differential regulation of these two CNS Na_v_ channel isoforms [7]. To that end, we employ the selective inhibitor of Wee1 kinase, called Wee1 Inhibitor II, and show that pharmacological inhibition of Wee1 kinase confers marked modulatory effects on FGF14:Na_v_1.2 complex assembly, but not FGF14:Na_v_1.6 complex assembly. Functionally, these effects of Wee1 inhibitor II are recapitulated, as pharmacological inhibition of Wee1 kinase alters FGF14-mediated regulation of the Na_v_1.2 channel, but not of the Na_v_1.6 channel. At the molecular level, these effects of Wee1 inhibitor II on FGF14:Na_v_1.2 complex assembly and the activity of the Na_v_1.2 channel macromolecular complex are shown to be dependent upon the presence of a residue of FGF14 previously shown to be prominent site for phosphorylation-mediated regulation of the protein [14]. Overall, these findings suggest that Wee1 kinase selectively modulates FGF14-mediated regulation of the Na_v_1.2 channel, which has important implications for understanding molecular mechanisms that fine-tune neuronal excitability.

## 2. Materials and Methods

### 2.1. Chemicals

D-luciferin (Gold Biotechnology, St. Louis, MO, USA) was prepared as a 30 mg/mL stock solution in phosphate-buffered saline (PBS), and stored at −20 °C. Wee1 inhibitor II (Calbiochem, San Diego, CA, USA) was reconstituted in 100% dimethyl sulfoxide (DMSO; Sigma-Aldrich, St. Louis, MO, USA) as 50 mM stock solutions and stored at −20 °C.

### 2.2. Plasmid Constructs

The following plasmid constructs used in this study were engineered and characterized as previously described [8,14,15,23,24,25,26,27,28]: CLuc-FGF14, CD4-Na_v_1.2 CTD-NLuc, CD4-Na_v_1.6 CTD-NLuc, GFP, FGF14-GFP, and FGF14^Y158A^-GFP. 

### 2.3. Cell Culture 

HEK293 cells were cultured as previously described [26,29,30,31], with different concentrations of G418 (Invitrogen, Carlsbad, CA, USA) added to the media to ensure stable Na_v_1.2 and Na_v_1.6 expression. 

### 2.4. Split-Luciferase Complementation Assay (LCA)

The split-lucifease complementation assay (LCA) was performed as previously described [15,23]. Briefy, HEK293 cells were transiently transfected with either the CLuc-FGF14 and CD4-Na_v_1.2 CTD-NLuc or CLuc-FGF14 and CD4-Na_v_1.6 CTD-NLuc pairs of DNA constructs using Lipofectamine 2000 (Invitrogen) according to the manufacturer’s instructions. 48 h post-transfection, transiently transfected cells were replated into CELLSTAR µClear^®^ 96-well tissue culture plates (Greiner Bio-One, Monroe, NC, USA). After 24 h, medium was replaced with serum-free, phenol-red free, 1:1 DMEM/F12 (Invitrogen) containing Wee1 inhibitor II (Calbiochem) dissolved in DMSO (1–150 µM) or DMSO alone. The final concentration of DMSO was maintained at 0.5% for all wells. Following 2 h incubation at 37 °C, the reporter reaction was initiated by addition of 100 µL substrate solution containing 1.5 mg/mL D-luciferin (Gold Biotechnologies) dissolved in PBS. Luminescence readings were then performed using a Synergy™ H1 Multi-Mode Microplate Reader (BioTek, Winooski, VT, USA). Acquired data was then analyzed as previously described [15,23].

### 2.5. Whole-Cell Voltage-Clamp Recordings in Heterologous Cells 

HEK-Na_v_1.2 or HEK-Na_v_1.6 cells were transiently transfected with pQBI-GFP, pQBI-FGF14-GFP, or pQBI-FGF14Y158A-GFP constructs. Twenty-four hours post-transfection, transiently transfected cells were plated at low density onto glass cover slips. After at least 2 h incubation, cover slips were transferred to a recording chamber containing extracellular recording solution comprised of the following salts: 140 mM NaCl; 3 mM KCl; 1 mM MgCl_2_; 1 mM CaCl_2_; 10 mM HEPES; and 10 mM glucose (final pH = 7.3; all salts purchased from Sigma-Aldrich). For control recordings, DMSO was added to the extracellular solution, whereas for recordings to characterize the effects of Wee1 inhibitor II (Calbiochem), the compound was added to the extracellular solution. The concentration of DMSO was maintained at 0.1% in both conditions. Cover slips were incubated for 30 min in extracellular solution containing vehicle or Wee inhibitor II (Calbiochem) prior to the start of recordings. For these recordings, borosilicate glass pipettes (Harvard Apparatus, Holliston, MA, USA) with a resistance of 3–5 MΩ, which were manufactured using a PC-100 vertical Micropipette Puller (Narishige International Inc., East Meadow, NY, USA), were filled with an intracellular solution comprised of the following salts: 130 mM CH_3_O_3_SCs; 1 mM EGTA; 10 mM NaCl; and 10 mM HEPES (pH = 7.3; all salts purchased from Sigma-Aldrich). After GΩ seal formation and entry into the whole-cell configuration, four voltage-clamp protocols were employed. The current–voltage (IV) protocol entailed voltage-steps from −100 mV to +60 mV from a holding potential of −70 mV. The voltage-dependence of steady-state inactivation protocol entailed a paired-pulse protocol during which, from the holding potential, cells were stepped to varying test potentials between −120 mV and +20 mV prior to a test pulse to −20 mV. For long-term inactivation, the voltage-clamp protocol entailed four depolarizations at 0 mV for 16 ms separated by three recovery intervals at −90 mV for 40 ms. For use-dependency, cells were stimulated using a train of 20 depolarization steps to −10 mV at a frequency of 10 Hz [32]. Recordings were performed using either an Axopatch 200B or Multiclamp 700B amplifier (Molecular Devices, Sunnyvale, CA, USA). Membrane capacitance and series resistance were estimated using the dial settings on the amplifier, and capacitive transients and series resistances were compensated by 70–80%. Data acquisition and filtering occurred at 20 and 5 kHz, respectively, before digitization and storage. Clampex 9.2 software (Molecular Devices) was used to set experimental parameters, and electrophysiological equipment was interfaced to this software using a Digidata 1200 analog–digital interface (Molecular Devices). Acquired data was then analyzed as previously described [27,28,30,31,33]. 

## 3. Results

### 3.1. Pharmacological Inhibition of Wee1 Kinase Modulates FGF14:Na_v_1.2, but Not FGF14:Na_v_1.6, Complex Assembly in a FGF14^Y158^-Dependent Manner 

To study the regulatory effects of Wee1 kinase on FGF14′s PPI with Na_v_ channel isoforms, we employed the pharmacological inhibitor of Wee1 kinase called Wee1 inhibitor II (Calbiochem). Wee1 inhibitor II (chemical name: 6-Butyl-4-(2-chlorophenyl)-9-hydroxypyrrolo[3, 4-c] carbazole-1,3(2H,6H)-dione), also referred to as compound 103 when first described [34], inhibits Wee1 kinase activity with an IC_50_ value of 59 nM [34]. Importantly, Wee1 inhibitor II displays ~590-fold selectivity over the related kinase Chk1, displaying an IC_50_ value of 35 µM toward the respective kinase [34]. Among Wee1 kinase inhibitors disclosed, these two combined features of Wee1 inhibitor II (i.e., IC_50_ values of 59 nM and 35 µM toward Wee1 and Chk1, respectively) conferred it with the best Chk1 IC_50_ to Wee1 IC_50_ ratio, marking it as currently the most targeted pharmacological inhibitor of Wee1 kinase activity [34].

Using an in-cell, split-luciferase complementation assay (LCA) previously optimized by our laboratory to identify modulators of the assembly of FGF:Na_v_ channel pairs [15,23], we tested the effects of pharmacological inhibition of Wee1 kinase on FGF14:Na_v_1.2 and FGF14:Na_v_1.6 complex assembly. In our studies, HEK293 cells were transiently transfected with either the CD4-Na_v_1.2 CTD-NLuc and CLuc-FGF14 cDNA constructs or the CD4-Na_v_1.6 CTD-NLuc and CLuc-FGF14 cDNA constructs. Resultantly, when FGF14 interacts with the CTD of the Na_v_1.2 or Na_v_1.6 channel, there is reconstitution of the NLuc and CLuc fragments of the luciferase enzyme, which, in the presence of the substrate luciferin, gives rise to a robust luminescence signal.

When tested for its effects on FGF14:Na_v_1.2 complex assembly, Wee1 inhibitor II gave rise to a dose-dependent decrease in FGF14:Na_v_1.2 complex assembly, as evidenced by the reduction in the luminescent signal observed in the presence of concentrations of Wee1 inhibitor II greater than or equal to 15 µM (Figure 1A). Plotting percentage luminescence as a function of the log concentration of Wee1 inhibitor II, the IC_50_ value of the pharmacological inhibitor of Wee1 kinase as it relates to decreasing FGF14:Na_v_1.2 complex assembly was determined to be 16.51 µM (Figure 1A).

Based upon the Y158 residue of FGF14 (FGF14^Y158^) serving as a prominent site for phosphorylation-mediated regulation of the protein [14], we next investigated if Wee1 kinase might exert its regulatory effects on FGF14:Na_v_1.2 complex assembly through a mechanism dependent upon the presence of the residue. When tested for its effects on FGF14:Na_v_1.2 complex assembly in conditions in which FGF14^WT^ was mutated to FGF14^Y158A^, Wee1 inhibitor II, even at the highest concentration tested (i.e., 150 µM), failed to reduce the luminescent signal by greater than 20% compared to per plate control wells treated with 0.5% DMSO (Figure 1B). These findings, in stark contrast to those observed in the WT condition where high concentrations of Wee1 inhibitor II conferred changes in the luminescent signal of greater than or equal to 80% (Figure 1A), provide strong evidence that the mechanism of action of pharmacological inhibition of Wee1 kinase as it relates to regulating FGF14:Na_v_1.2 complex assembly is dependent upon the presence of FGF14^Y158^.

Given that FGF14 confers differential regulation of the Na_v_1.2 and Na_v_1.6 channels [7], we next investigated if there were potentially different effects of pharmacological inhibition of Wee1 kinase on FGF14:Na_v_1.2 versus FGF14:Na_v_1.6 complex assembly. In stark contrast to the robust modulatory effects on FGF14:Na_v_1.2 complex assembly conferred by Wee1 inhibitor II, pharmacological inhibition of Wee1 kinase exerted no noticeable effects on FGF14:Na_v_1.6 complex assembly (Figure 1C). Overall, these data suggest that Wee1 kinase might exert selective regulation of the Na_v_1.2 channel macromolecular complex.

### 3.2. Pharmacological Inhibition of Wee1 Kinase Modulates FGF14-Mediated Regulation of Na_v_1.2 Channels

Having shown that pharmacological inhibition of Wee1 kinase confers modulatory effects on FGF14:Na_v_1.2 complex assembly (Figure 1A), we next sought to investigate if these alterations in complex assembly resulted in changes in the function of the Na_v_1.2 channel. To do so, HEK293 cells stably expressing the Na_v_1.2 channel (HEK-Na_v_1.2) [26,29,30] were transiently transfected with either GFP (HEK-Na_v_1.2-GFP) or FGF14-GFP (HEK-Na_v_1.2-FGF14-GFP). Cells were then incubated for 30 min with either vehicle (0.1% DMSO) or 15 µM of Wee1 inhibitor II, a concentration selected on the basis of it being close to the IC_50_ value of the compound as determined in the LCA (Figure 1A). After incubation, whole-cell voltage-clamp recordings were performed to characterize the effects of pharmacological inhibition of Wee1 kinase on the activity of Na_v_1.2 channels (Figure 2; Table 1 and Table 2).

Consistent with previous investigations [7], co-expression of FGF14-1b, the isoform of FGF14 studied in this investigation, with the Na_v_1.2 channel α subunit in heterologous leads to a reduction in Na_v_1.2-mediated peak *I*_Na_ density (−128.7 ± 5.7 pA/pF (*n* = 11) and −89.24 ± 7.7 pA/pF (*n* = 11) for HEK-Na_v_1.2-GFP + DMSO and HEK-Na_v_1.2-FGF14-GFP + DMSO, respectively; Figure 2A–C). Notably, whereas Wee1 inhibitor II displayed no effects on Na_v_1.2-mediated *I*_Na_ in the absence of FGF14, pharmacological inhibition of Wee1 kinase led to an exacerbation of FGF14-mediated suppression of Na_v_1.2-mediated *I*_Na_, as evidenced by HEK-Na_v_1.2-FGF14-GFP cells treated with Wee1 inhibitor II displaying an average peak *I*_Na_ density (−29.6 ± 7.29 pA/pF; *n* = 9) significantly less than HEK-Na_v_1.2-FGF14-GFP cells treated with vehicle (−89.24 ± 7.7 pA/pF; *n* = 11; Figure 2A–C). This effect of Wee1 inhibitor II on Na_v_1.2-mediated peak *I*_Na_ density is similar to the effect of pharmacological inhibition of Wee1 kinase on tau of fast inactivation of Na_v_1.2-mediated *I*_Na_. Namely, co-expression of FGF14 with the Na_v_1.2 channel leads to a slowing of the entry of Na_v_1.2 channels into fast inactivation, as evidenced by the increased tau value observed between the HEK-Na_v_1.2-GFP + DMSO (0.88 ± 0.10 ms; *n* = 11) and HEK-Na_v_1.2-FGF14-GFP + DMSO (1.16 ± 0.08 ms; *n* = 11) groups, and pharmacological inhibition of Wee1 kinase exacerbates this FGF14-mediated regulatory effects, with the average tau value observed in the HEK-Na_v_1.2-FGF14-GFP + Wee1 inhibitor II group (2.13 ± 0.35 ms; *n* = 9) being significantly greater than that observed in the HEK-Na_v_1.2-FGF14-GFP + DMSO group (1.16 ± 0.08 ms; *n* = 11; Figure 2D).

Consistent with previous investigations [7], co-expression of FGF14 with the Na_v_1.2 channel led to a depolarizing shift in the voltage-dependence of Na_v_1.2 channel activation (V_1/2_ of activation = −26.61 ± 1.1 mV (*n* = 11) and −21.09 ± 1.2 mV (*n* = 11) for HEK-Na_v_1.2-GFP + DMSO and HEK-Na_v_1.2-FGF14-GFP + DMSO, respectively; Figure 2E,F). Similar to the effects of Wee1 inhibitor II on peak *I*_Na_ density and tau of fast inactivation, pharmacological inhibition of Wee1 displays no effect in the absence of FGF14, but exacerbates FGF14’s regulatory effects on the voltage-dependence of Na_v_1.2 channel activation, with the average V_1/2_ of activation value observed in the HEK-Na_v_1.2-FGF14-GFP + Wee1 inhibitor II group (−15.51 ± 0.9 mV; *n* = 8) being significantly more depolarized relative to that observed in the HEK-Na_v_1.2-FGF14-GFP + DMSO group (−21.09 ± 1.2 mV; *n* = 11; Figure 2E,F).

Whereas Wee1 inhibitor II exacerbates FGF14-mediated regulation of peak *I*_Na_ density, tau of fast inactivation, and the voltage-dependence of activation of Na_v_1.2 channels in heterologous cells, pharmacological inhibition of Wee1 kinase also results in FGF14 displaying entirely altered regulatory on other biophysical properties. Specifically, FGF14 does not inherently modulate the voltage-dependence of Na_v_1.2 channel steady-state inactivation (V_1/2_ of steady-state inactivation = −59.32 ± 0.7 mV (*n* = 8) and −67.76 ± 5.2 mV (*n* = 8) for HEK-Na_v_1.2-GFP + DMSO and HEK-Na_v_1.2-FGF14-GFP + DMSO, respectively; Figure 2G,H). However, when Wee1 kinase is pharmacologically inhibited, there is an apparent depolarizing shift in the voltage-dependence of Na_v_1.2 channel steady-state inactivation observed in the presence of FGF14, as evidenced by the ~12 mV depolarizing shift in this parameter between the HEK-Na_v_1.2-FGF14-GFP + DMSO (−67.76 ± 5.2 mV; *n* = 8) and HEK-Na_v_1.2-FGF14-GFP + Wee1 inhibitor II (−55.14 ± 1.1 mV; *n* = 7) groups (Figure 2G,H). Notably, no effect of Wee1 inhibitor II is observed on Na_v_1.2 steady-state inactivation in the absence of FGF14 (Figure 2G,H), suggesting that this effect is dependent upon the presence of FGF14.

This effect of Wee1 inhibitor II as it relates to entirely altering the function of FGF14 is also evident when investigating how pharmacological inhibition of Wee1 kinase affects long-term and cumulative inactivation of Na_v_1.2 channels. Specifically, FGF14 does not inherently affect the fraction of Na_v_1.2 channels that enter into long-term inactivation (Figure 2I; Table 2). Intriguingly, however, HEK-Na_v_1.2-FGF14-GFP cells treated with Wee1 inhibitor II display a peculiar phenotype in which the *I*_Na_ amplitude observed during the 2nd, 3rd, and 4th depolarization cycles is significantly larger than that observed during the 1st depolarization cycle, as evidenced by the *I*_Na_ ratio during the 2nd, 3rd, and 4th depolarization cycles being greater than 100% (Figure 2I; Table 2). Relatedly, FGF14 does not inherently affect the fraction of Na_v_1.2 channels that undergo cumulative inactivation; however, treatment of HEK-Na_v_1.2-FGF14-GFP cells with Wee1 kinase inhibitor II results in a phenotype in which the *I*_Na_ amplitude observed after the 20th pulse is larger than that observed during the 1st pulse (Figure 2J,K). Notably, these effects observed during repetitive stimulation are only present in the HEK-Na_v_1.2-FGF14-GFP + Wee1 inhibitor II group (Figure 2I–K), suggesting that the phenotype is dependent upon interplay between Wee1 kinase inhibition and FGF14.

### 3.3. Effects of Wee1 Inhibitor II on FGF14-Mediated Regulation of Na_v_1.2 Channels Is Depedent upon the Presence of FGF14^Y158^

Previously, we showed that through phosphoryation of FGF14^Y158^, the tyrosine kinase JAK2 modulates FGF14-mediated regulation of Na_v_ channel activity [14]. As Wee1 is also a tyrosine kinase, and given the robust modulatory effects of pharmacological inhibition of Wee1 on FGF14-mediated regulation of the Na_v_1.2 channel shown in Figure 2, we next investigated if FGF14^Y158^ was also necessary for Wee1 inhibitor II to exert its regulatory effects on FGF14 activity. To that end, HEK-Na_v_1.2 cells were transiently transfected with FGF14^Y158A^-GFP (HEK-Na_v_1.2-FGF14^Y158A^-GFP), and the effects of Wee1 inhibitor II on currents mediated by these cells were assessed using the voltage-clamp protocols employed in Figure 2.

Whereas Wee1 inhibitor II exacerbated FGF14-mediated suppression of Na_v_1.2-mediated peak *I*_Na_ density in HEK-Na_v_1.2-FGF14-GFP cells (Figure 2A–C), pharmacological inhibition of Wee1 kinase had no effect on this parameter when tested in HEK-Na_v_1.2-FGF14^Y158A^-GFP cells (Figure 3A–C). Relatedly, whereas Wee1 inhibitor II exacerbated FGF14′s effects on tau of fast inactivation (Figure 2D) and the voltage-dependence of activation (Figure 2E,F) of Na_v_1.2 channels in HEK-Na_v_1.2-FGF14-GFP cells, the compound did not alter either of these parameters in HEK-Na_v_1.2-FGF14^Y158A^-GFP cells (Figure 3D–F). Additionally, whereas Wee1 inhibitor II altered the function of FGF14 in HEK-Na_v_1.2-FGF14-GFP cells and induced a depolarizing shift in the voltage-dependence of Na_v_1.2 channel steady-state inactivation in the presence of FGF14 (Figure 2G,H), pharmacological inhibition of Wee1 kinase had no effects on this parameter in the presence of FGF14^Y158A^ (Figure 3G,H).

In the presence of FGF14, Wee1 inhibitor II also exerted a peculiar effect on the long-term and cumulative inactivation of Na_v_1.2 channels that was characterized by an increase in the amplitude of *I*_Na_ after repetitive stimulation (Figure 2I–K). In HEK-Na_v_1.2-FGF14^Y158A^-GFP cells, however, the ratio of *I*_Na_ amplitude (normalized to the *I*_Na_ amplitude observed during the first depolarization) of the 2nd, 3rd, and 4th depolarizations cycles was not different in the long-term inactivation protocol when comparing HEK-Na_v_1.2-FGF14^Y158A^-GFP cells treated with vehicle to HEK-Na_v_1.2-FGF14^Y158A^-GFP cells treated with Wee1 inhibitor II (Figure 3I). Relatedly, the ratio of *I*_Na_ amplitude (normalized to the *I*_Na_ amplitude observed during the first depolarization) was not different for any of the depolarization cycles in the cumulative inactivation protocol for HEK-Na_v_1.2-FGF14^Y158A^-GFP cells treated with vehicle versus HEK-Na_v_1.2-FGF14^Y158A^-GFP cells treated with Wee1 inhibitor II (Figure 3J,K). Interestingly, however, HEK-Na_v_1.2-FGF14^Y158A^-GFP cells treated with both vehicle and Wee1 inhibitor II displayed larger *I*_Na_ amplitudes after repetitive stimulation compared to their *I*_Na_ amplitudes observed after the first depolarization cycle (Figure 3K).

### 3.4. Wee1 Inhibitor II Does Not Affect FGF14-Mediated Regulation of the Na_v_1.6 Channel

Having shown in our LCA experiments that Wee1 inhibitor II did not affect FGF14:Na_v_1.6 complex assembly (Figure 1C), we next sought to investigate if, and unlike the modulatory of Wee1 inhibitor II on FGF14-mediated regulatory effects of the Na_v_1.2 channel (Figure 2), this would preclude the ligand from conferring functional modulation of the Na_v_1.6 channel or FGF14:Na_v_1.6 complex. To do so, HEK293 cells stably expressing the Na_v_1.6 channel (HEK-Na_v_1.6) were transiently transfected with either GFP (HEK-Na_v_1.6-GFP) or FGF14-GFP (HEK-Na_v_1.6-FGF14-GFP). Cells were incubated for 30 min prior to recording with either vehicle (0.1% DMSO) or 15 µM Wee1 inhibitor II, and the voltage-clamp protocols employed in Figure 2 and Figure 3 were used to characterize effects of pharmacological inhibition of Wee1 kinase on Na_v_1.6 channel activity (Figure 4; Table 3 and Table 4).

Consistent with the results of previous investigations [14,25,26,27,28,30], co-expression of FGF14 with the Na_v_1.6 channel in heterologous cells lead to a reduction in the peak *I*_Na_ density of Na_v_1.6-mediated currents, as evidenced by HEK-Na_v_1.6-FGF14-GFP cells displaying an average peak *I*_Na_ density (−17.31 ± 2.5 pA/pF; *n* = 10) significantly less than HEK-Na_v_1.6-GFP cells (−57.83 ± 6.3 pA/pF; *n* = 8; Figure 4A–C). Whereas Wee1 inhibitor II exacerbated this FGF14-mediated regulatory on Na_v_1.2-mediated currents (Figure 2A–C), pharmacological inhibition of Wee1 kinase did not affect Na_v_1.6-mediated peak *I*_Na_ density in the absence or presence of FGF14 (Figure 4A–C).

Consistent with previous investigations [14,25,26,27,28,30], co-expression of FGF14 with the Na_v_1.6 channel lead to a slowing of Na_v_1.6 channel fast inactivation (Figure 4D) and a depolarizing shift in the voltage-dependence of Na_v_1.6 channel activation (Figure 4E,F). These findings are evidenced by HEK-Na_v_1.6-GFP cells displaying a tau of fast inactivation of 1.03 ± 0.04 ms (*n =* 8), whereas HEK-Na_v_1.6-FGF14-GFP cells display a significantly larger value for this parameter of 1.64 ± 0.26 ms (*n* = 8; Figure 4D), and HEK-Na_v_1.6-GFP cells displaying a V_1/2_ of activation of −22.87 ± 1.69 mV (*n* = 8), whereas HEK-Na_v_1.6-FGF14-GFP cells display a significantly more depolarized V_1/2_ of activation of −18.15 ± 1.03 mV (*n* = 8; Figure 4E,F), respectively. Whereas Wee1 inhibitor II exacerbated FGF14-mediated regulatory effects on the tau of fast inactivation and the voltage-dependence of activation of Na_v_1.2 channels (Figure 2D–F), pharmacological inhibition of Wee1 kinase did not affect the tau of fast of inactivation or the voltage-dependence of activation of Na_v_1.6 channels in the absence of presence of FGF14 (Figure 4D–F). Relatedly, whereas Wee1 inhibitor II entirely altered the regulatory effects of FGF14 on steady-state inactivation, namely, inducing a depolarizing shift in the V_1/2_ of Na_v_1.2 steady-state inactivation despite FGF14 not inherently regulating this parameter of Na_v_1.2 channels (Figure 2G,H), pharmacological inhibition of Wee1 kinase did not affect the voltage-dependence of Na_v_1.6 steady-state inactivation in the absence or presence of FGF14 (Figure 4G,H). This finding is particularly notable on account of FGF14′s differential regulation of the voltage-dependence of steady-state inactivation of Na_v_1.2 versus Na_v_1.6, where FGF14 does not modulate this parameter of Na_v_1.2 channels (Figure 2G,H), but confers a depolarizing shift in the voltage-dependence of Na_v_1.6 channel steady state-inactivation (Figure 4G,H) when the two proteins are co-expressed in heterologous cells, which is a finding consistent with previous investigations [7].

As also observed in previous studies [14,30], co-expression of FGF14 with the Na_v_1.6 channel in heterologous cells resulted in a reduction in the fraction of Na_v_1.6 channels entering into long-term inactivation (Figure 4I). This is evidenced by the *I*_Na_ ratio (normalized to the *I*_Na_ amplitude during the first depolarization) of 2nd, 3rd, and 4th depolarization cycles being greater in HEK-Na_v_1.6-FGF14-GFP cells compared to HEK-Na_v_1.6-GFP cells (Figure 4I; Table 4). However, Wee1 inhibitor II did not affect the entry of Na_v_1.6 channels into long-term inactivation in the absence or presence of FGF14 (Figure 4I; Table 4). Similarly, pharmacological inhibition of Wee1 kinase did not affect the cumulative inactivation of Na_v_1.6 channels in the absence or presence of FGF14 (Figure 4J,K).

## 4. Discussion

Intracellular fibroblast growth factors (iFGF; FGF11-FGF14) have emerged as important Na_v_ channel auxiliary proteins [6,7,8,10,35,36,37,38,39,40]. Notably, PPIs between different iFGFs and different Na_v_ channel isoforms contribute to the generation of phenotypically distinct sodium currents with specialized functions in different tissues [7,9,37,41,42,43]. For example, the PPI between FGF12 and the Na_v_1.5 channel confers important regulatory effects on the *I*_Na_ of cardiomyocytes [41,42]; the PPI between FGF13 and the Na_v_1.7 channel confers important regulatory effects on the *I*_Na_ of dorsal root ganglion neurons [9,43,44]; and the PPI between FGF14 and the Na_v_1.2 and Na_v_1.6 channel is important for regulating the *I*_Na_ and excitability of hippocampal and striatal neurons [7,14,26,37,45]. Despite these diverse and well-established regulatory effects of iFGFs on Na_v_ channels, less is known about cellular signaling molecules that might regulate and enable the targeted modulation conferred by different iFGFs on different Na_v_ channel isoforms. To that end, and focusing on the PPI between FGF14 and the Na_v_1.2 channel and Na_v_1.6 channel given the primacy of these PPIs in regulating neuronal Na+ currents and excitability [7,11,12,14,26,27,37,45], we built upon previous studies suggesting a potential role of Wee1 kinase in modulating FGF14-mediated regulation of central nervous system Na_v_ channel isoforms [13,14,17].

In our previous studies investigating kinase networks that might regulate Na_v_ channel macromolecular complexes, evidence emerged of an intricate pathway involving GSK3, AKT, and Wee1 that might regulate Na_v_ channel activity and neuronal excitability [13,17]. Supporting such a hypothesis, we have previously shown that GSK3-mediated phosphorylation of T1966 of Na_v_1.2 and T1936 of Na_v_1.6 confers important regulatory effects on the biophysical properties of these channels and on the excitability of neurons in clinically relevant brain regions [16,29]. Additionally, we have shown that AKT is an important regulator of repetitive firing of hippocampal and striatal neurons [16,17]. Given that Wee1 kinase is regulated and degraded by GSK3 through ubiquitination [19,20,21], and that Wee1 kinase may increase the activity of AKT [22], which would be predicted to, in turn, inhibit GSK3 activity [17], we sought in this work to take a focused approach toward unraveling the complex role of Wee1 in regulating this kinase network.

In a previous study, Wee1 inhibitor II was identified as the most targeted pharmacological inhibitor of Wee1 kinase [34]. Using this chemical tool, we tested the effects of pharmacological inhibition of Wee1 kinase in our previously described LCA optimized to identify modulators of complex’s involving iFGFs and Na_v_ channels [15,23]. When tested for modulatory effects on FGF14:Na_v_1.2 complex assembly (Figure 1A), Wee1 inhibitor II was shown to demonstrate dose-dependent inhibition of the complex’s formation. This finding could suggest that Wee1 kinase phosphorylates residues at the FGF14:Na_v_1.2 PPI interface that are important for complex assembly. Supporting such a hypothesis, when the effects of Wee1 inhibitor II on FGF14:Na_v_1.2 complex assembly were assayed in conditions featuring FGF14^Y158A^, which represents an FGF14 point mutation previously shown to abrogate phosphorylation-mediated regulation of FGF14 by JAK2 [14], the compound failed to exert noticeable effects of FGF14:Na_v_1.2 complex assembly (Figure 1B). This finding could suggest that Wee1 kinase exerts its modulatory effects on FGF14:Na_v_1.2 complex assembly through phosphorylation of FGF14^Y158^; however, future in vitro phosphorylation assays that demonstrate phosphorylation of FGF14^Y158^ by Wee1 kinase are necessary to unequivocally substantiate such a hypothesis.

In contrast to the modulatory effects on FGF14:Na_v_1.2 complex assembly exerted by Wee1 inhibitor II, the compound did not exert noticeable effects on FGF14:Na_v_1.6 complex assembly in our LCA experiments (Figure 1C). This lack of appreciable effects of Wee1 inhibitor II on FGF14:Na_v_1.6 complex assembly is consistent with previous investigations [13]. However, in our previous study, Wee1 inhibitor I (chemical name: 4-(2-Chlorophenyl)−9-hydroxypyrrolo[3,4-c]carbazole-1,3(2H,6H)-dione; referred to as compound 23 when initially disclosed [34]) did exert appreciable inhibitory effects on FGF14:Na_v_1.6 complex assembly at high micromolar concentrations [13]. Given that Wee1 inhibitor I was previously shown to be less selective for kinases related to Wee1 kinase compared to Wee1 inhibitor II [34], these differential effects of the two compounds on FGF14:Na_v_1.6 complex assembly are presumed to arise due to the different selectivity profiles of the two compounds, with the inhibitory effects of Wee1 inhibitor I conferred on FGF14:Na_v_1.6 complex assembly presumed to be due to off-target modulation of other kinases.

Having shown that Wee1 inhibitor II exerts robust modulatory effects on FGF14:Na_v_1.2 complex assembly, we next sought to investigate if pharmacological inhibition of Wee1 kinase would alter FGF14-mediated regulation of the Na_v_1.2 channel. In support of Wee1 kinase conferring modulatory effects on FGF14-mediated regulation of the Na_v_1.2 channel, Wee1 inhibitor II displayed effects on Na_v_1.2 mediated currents, but only in the presence of FGF14. In particular, pharmacological inhibition of Wee1 kinase was shown to exacerbate FGF14-mediated regulatory effects on some electrophysiological parameters of the Na_v_1.2 channel (i.e., peak *I*_Na_ density, tau of fast inactivation, and voltage-dependence of activation) and entirely alter FGF14 activity with respect to other electrophysiological parameters. As it pertains to the latter, Wee1 inhibitor II, in the presence of FGF14, lead to a depolarizing shift in the voltage-dependence of Na_v_1.2 channel steady-state inactivation, an electrophysiological parameter that is not inherently modulated by FGF14 (Figure 2G,H). Likewise, pharmacological inhibition of Wee1 kinase resulted in an increased *I*_Na_ amplitude after repetitive simulation (when compared to the *I*_Na_ amplitude observed during the first depolarization cycle of either the long-term inactivation or cumulative inactivation protocols; Figure 2I–K), which are also not regulatory effects inherent to FGF14. Collectively, these findings suggest that Wee1 kinase confers complex regulatory effects on the Na_v_1.2 channel by altering the activity of FGF14.

To elucidate molecular determinants of how Wee1 kinase alters FGF14-mediated regulation of the Na_v_1.2 channel, and based upon the LCA data shown in Figure 1B, we tested Wee1 inhibitor II in HEK-Na_v_1.2 cells co-expressing FGF14^Y158A^ (Figure 3). In contrast to the myriad of effects of Wee1 inhibitor II on FGF14-mediated regulation of the Na_v_1.2 channel in the presence of WT FGF14 (Figure 2), pharmacological inhibition of Wee1 kinase did not confer any changes in FGF14-mediated regulation of the Na_v_1.2 channel in the presence of FGF14^Y158A^. Coupled with the LCA data shown in Figure 1B, these functional data provide strong evidence for FGF14^Y158^ being an important residue that confers Wee1 with its regulatory effects on the Na_v_1.2 channel macromolecular complex.

Given that FGF14 has previously been shown to confer differential modulation of the Na_v_1.2 and Na_v_1.6 channels, coupled with the lack of effect of Wee1 inhibitor II on FGF14:Na_v_1.6 complex assembly in the LCA experiments (Figure 1C), we lastly investigated if pharmacological inhibition of Wee1 kinase might confer different functional effects on FGF14-mediated regulation of the Na_v_1.2 channel versus Na_v_1.6 channel. Consistent with the LCA experiments showing no effects of Wee1 inhibitor II on FGF14:Na_v_1.6 complex assembly (Figure 1C), pharmacological inhibition of Wee1 kinase did not affect Na_v_1.6-mediated Na+ currents in the absence or presence of FGF14 (Figure 4). Collectively considered, these data suggest that there could be residues at the FGF14:Na_v_1.2 PPI interface that are targeted by Wee1 kinase that are not conserved among other iFGF:Na_v_ channel pairs; although, extensive structural and biophysical investigations would be necessary to unequivocally substantiate such suppositions.

Overall, by complementarily employing the LCA and whole-cell patch-clamp electrophysiology in heterologous cells, we have demonstrated that pharmacological inhibition of Wee1 kinase confers modulatory effects on the Na_v_1.2, but not the Na_v_1.6, macromolecular complex. As both of these Na_v_ channel isoforms are enriched in clinically relevant brain regions with different subcellular distributions [46], our findings have important implications for understanding cellular signaling molecules that fine-tune neuronal excitability.

In particular, and given the selective effects of Wee1 inhibitor II on FGF14-mediated regulation of the Na_v_1.2 channel, but not the Na_v_1.6 channel, our data suggest that Wee1 kinase could confer targeted subcellular regulation of neuronal activity. Such a hypothesis is supported by Na_v_1.2 channels being enriched in the somatodendritic region and proximal axon initial segment of neurons where they contribute to action potential backpropagation and spike timing-dependent plasticity, whereas Na_v_1.6 channels are enriched in the distal region of the axon initial segment where they contribute to the forward propagation of action potentials and repetitive firing [46,47,48,49]. As such, our data suggest that Wee1 kinase might have an important role in promoting action potential backpropagation and synaptic signal integration, although, future ex vivo current-clamp recordings would be necessary to validate such regulatory mechanisms [47].

## Figures and Tables

**Figure 1 cells-10-03103-f001:**
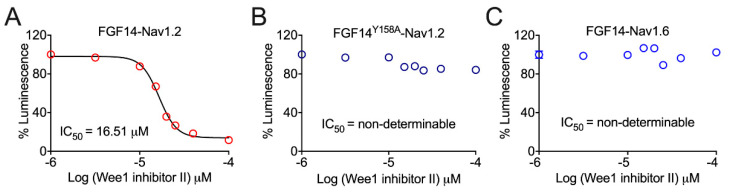
Evaluation of the effects of Wee1 inhibitor II on FGF14:Na_v_1.2 and FGF14:Na_v_1.6 complex assembly. (**A**) Percentage luminescence (normalized to per plate control wells treated with 0.5% DMSO; *n* = 32 wells per plate) plotted as a function of log concentration of Wee1 inhibitor II to characterize dose-dependent effects of pharmacological inhibition of Wee1 kinase on FGF14:Na_v_1.2 complex assembly (range of concentrations tested = 1 µM–150 µm; *n* = 6 wells per concentration). (**B**) Effects of different concentrations (range = 1 µM–150 µM; *n* = 6 wells per concentration) of Wee1 inhibitor II on FGF14^Y158A^:Na_v_1.2 complex assembly. (**C**) Effects of different concentrations of Wee1 inhibitor II (range = 1 µM–150 µM; *n* = 6 wells per concentration) on FGF14:Na_v_1.6 complex assembly. Data are represented as mean ± SEM.

**Figure 2 cells-10-03103-f002:**
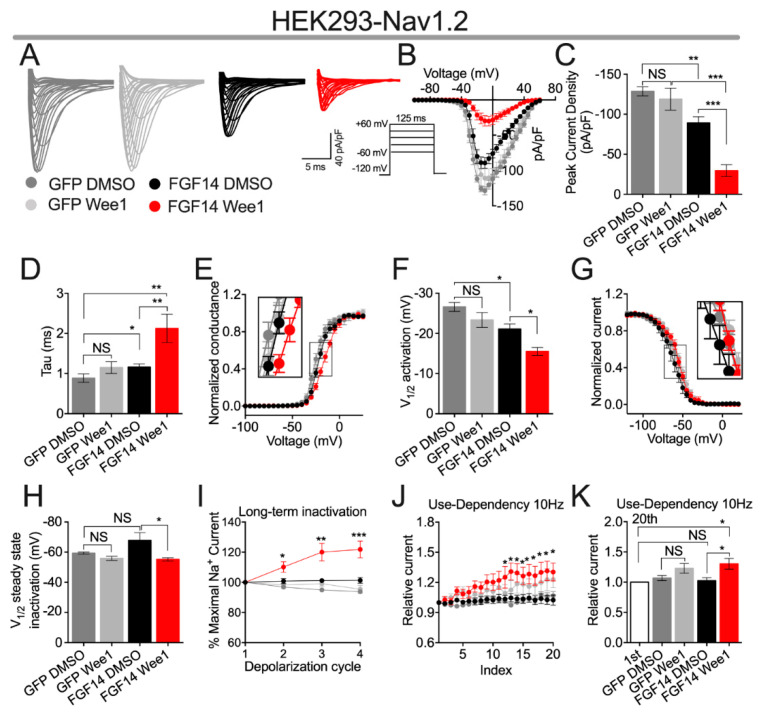
Functional evaluation of the effects of Wee1 inhibitor II on Na_v_1.2-mediated currents. (**A**) Representative traces of *I*_Na_ from cells of the indicated experimental groups in the response to the depicted voltage-clamp protocol. (**B**) Current–voltage relationships of cells from the experimental groups described in (**A**). (**C**,**D**) Comparison of peak *I*_Na_ density (**C**) and the tau of fast inactivation of *I*_Na_ (**D**) of cells from the indicated experimental groups. (**E**) Normalized conductance plotted as a function of voltage to characterize the voltage-dependence of Na_v_1.2 channel activation of cells from the experimental groups described in (**A**). Data were fitted with the Boltzmann equation to determine V_1/2_ of activation. (**F**) Bar graph derived from (**E**) comparing V_1/2_ of activation among the indicated experimental groups. (**G**) Normalized current plotted as a function of voltage to characterize the voltage-dependence of Na_v_1.2 channel steady-state inactivation of cells from the experimental groups described in (**A**). Data were fitted with the Boltzmann equation to determine V_1/2_ of steady-state inactivation. (**H**) Bar graph derived from (**G**) comparing V_1/2_ of Na_v_1.2 channel steady-state inactivation among the indicated experimental groups. (**I**) Percentage maximal *I*_Na_ (normalized to the *I*_Na_ amplitude observed during the first depolarization) plotted as a function of depolarization cycle to characterize the effects of Wee1 inhibitor II on the entry of Na_v_1.2 channels into long-term inactivation in the experimental groups described in (**A**). (**J**) Relative current (normalized to the *I*_Na_ amplitude observed during the first depolarization) plotted as a function of depolarization cycle to characterize the effects of pharmacological inhibition of Wee1 kinase on the cumulative inactivation of Na_v_1.2 channels for the experimental groups described in (**A**). (**K**) Bar graph comparing the ratio of the *I*_Na_ amplitude observed during the first depolarization normalized to the *I*_Na_ amplitude observed during the 20th depolarization for the indicated experimental groups. Data are mean ± SEM. Significance was assessed using a one-way ANOVA with post-hoc Tukey’s multiple comparisons test. *, *p* < 0.05; **, *p* < 0.01; ***_,_
*p* < 0.001. Data are summarized in Table 1 and Table 2.

**Figure 3 cells-10-03103-f003:**
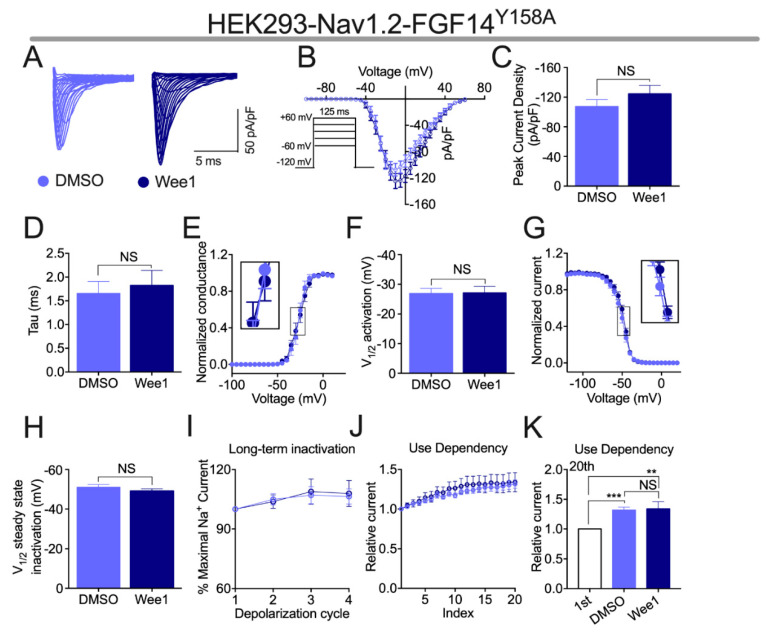
Functional evaluation of the effects of Wee1 inhibitor II in HEK-Na_v_1.2 cells co-expressing FGF14^Y158A^. (**A**) Representative traces of *I*_Na_ elicited by cells from the indicated experimental groups in response to the depicted voltage-clamp protocol. (**B**) Current–voltage relationship of cells from the experimental groups described in (**A**). (**C**) Comparison of the peak *I*_Na_ density of cells of the indicated experimental groups. (**D**) Comparison of tau of Na_v_1.2 channel fast inactivation between the indicated experimental groups. (**E**) Conductance–voltage relationship of Na_v_1.2 channels in the experimental groups described in (**A**). (**F**) Comparison of V_1/2_ of Na_v_1.2 channel activation between the indicated experimental groups. (**G**) Normalized current plotted as a function of voltage to characterize the voltage-dependence of Na_v_1.2 channel steady-state inactivation for the experimental groups described in (**A**). (**H**) Comparison of V_1/2_ of Na_v_1.2 channel steady-state inactivation between the indicated experimental groups. (**I**,**J**) Characterization of long-term inactivation (**I**) and cumulative inactivation (**J**) of Na_v_1.2 channels for the experimental groups described in (**A**). (**K**) Comparison of the relative *I*_Na_ amplitude at the 1st pulse to the 20th pulse for the indicated experimental groups. Data are mean ± SEM. Significance was assessed using a one-way ANOVA with post hoc Tukey’s multiple comparisons test. **, *p* < 0.01; ***, *p* < 0.001. Data are summarized in Table 1 and Table 2.

**Figure 4 cells-10-03103-f004:**
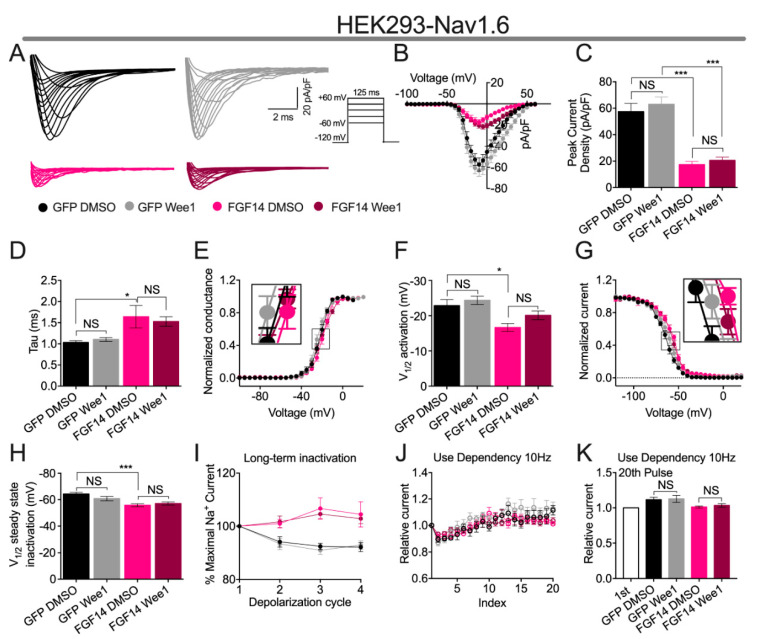
Wee1 inhibitor II does not affect FGF14-mediated regulation of the Na_v_1.6 channel. (**A**) Representative traces of *I*_Na_ elicited by cells of the indicated experimental groups in response to the depicted voltage-clamp protocol. (**B**) Current–voltage relationships of cells from the experimental groups described in (**A**). (**C**) Peak *I*_Na_ density of cells from the indicated experimental groups. (**D**) Tau of fast inactivation of cells from the indicated experimental groups. (**E**) Voltage-dependence of activation of cells from the experimental groups described in (**A**). (**F**) V_1/2_ of activation of cells from the indicated experimental groups. (**G**) Steady-state inactivation plots of cells from the experimental groups described in (**A**). (**H**) V_1/2_ of steady-state inactivation of cells from the indicated experimental groups. (**I**) Characterization of entry of Na_v_1.6 channels into long-term inactivation for the experimental groups described in (**A**). (**J**) Characterization of cumulative inactivation of Na_v_1.6 channels for the experimental groups described in (**A**). (**K**) Relative current at the 20th pulse (normalized to the *I*_Na_ amplitude observed during the first depolarization) for the indicated experimental groups. Data are mean ± SEM. Significance was assessed using a one-way ANOVA with post hoc Tukey’s multiple comparisons test. *, *p* < 0.05; ***, *p* < 0.001. Data are summarized in Table 3 and Table 4.

**Table 1 cells-10-03103-t001:** Effects of pharmacological inhibition of Wee1 kinase on Na_v_1.2-mediated currents ^†^.

Condition	Peak Density	Activation	K_act_	Steady-State Inactivation	K_inact_	Tau (τ)
	pA/pF	mV	mV	mV	mV	ms
GFP DMSO	−128.7 ± 5.7 (11)	−26.61 ± 1.1 (11)	4.22 ± 0.4 (11)	−59.32 ± 0.7 (8)	5.17 ± 0.28 (8)	0.88 ± 0.10 (11)
GFP Wee1	−118.8 ± 13.7 (9) ^ns^	−23.34 ± 1.8 (9)	4.50 ± 0.3 (9)	−55.92 ± 1.4 (10)	5.38 ± 0.29 (10)	1.15 ± 0.14 (10)
FGF14 DMSO	−89.24 ± 7.7 (11) ^a^	−21.09 ± 1.2 (11) ^d^	4.47 ± 0.5 (11)	−67.76 ± 5.2 (8)	6.04 ± 0.81 (8)	1.16 ± 0.08 (11) ^h^
FGF14 Wee1	−29.6 ± 7.29 (9) ^b,c^	−15.51 ± 0.9 (8) ^e^	6.58 ± 0.8 (8) ^f^	−55.14 ± 1.1 (7) ^g^	7.58 ± 0.61 (7)	2.13 ± 0.35 (9) ^i^
FGF14^Y158A^ DMSO	−107.3 ± 9.52 (10)	−26.88 ± 1.7 (10)	3.04 ± 0.3 (10)	−51 ± 1.4 (9)	5.76 ± 0.69 (9)	1.65 ± 0.25 (10)
FGF14^Y158A^ Wee1	−124.5 ± 11.4 (11) ^ns^	−27.08 ± 2.3 (11)	3.74 ± 0.4 (11)	−49.07 ± 1.2 (11)	5.35 ± 0.50 (10)	1.82 ± 0.32 (11)

^†^ Data are mean ± SEM; ns = nonsignificant; (*n*) = number of cells. ^a^ *p* = 0.010, One way ANOVA Tukey’s multiple comparisons test compared to GFP DMSO; ^b^ *p* = 0.0020, One way ANOVA Tukey’s multiple comparisons test compared to FGF14 DMSO; ^c^ *p* < 0.0001, One way ANOVA Tukey’s multiple comparisons test compared to GFP DMSO; ^d^ *p* = 0.029, One way ANOVA Tukey’s multiple comparisons test compared to GFP DMSO; ^e^ *p* = 0.048, One way ANOVA Tukey’s multiple comparisons test compared to FGF14 DMSO; ^f^
*p =* 0.020, One way ANOVA Tukey’s multiple comparisons test compared to FGF14 DMSO; ^g^ *p* = 0.020, One way ANOVA Tukey’s multiple comparisons test compared to FGF14 DMSO; ^h^ *p =* 0.047, unpaired *t* tests compared to GFP DMSO; ^i^ *p* = 0.0042, One way ANOVA Tukey’s multiple comparisons test compared to FGF14 DMSO.

**Table 2 cells-10-03103-t002:** Effects of pharmacological inhibition of Wee1 kinase on the entry of Na_v_1.2 channels into long-term inactivation ^†^.

Condition	LTI (% Maximal Na^+^ Current)		
	2nd Pulse	3rd Pulse	4th Pulse
GFP DMSO	96.87 ± 0.7 (7)	94.95 ± 0.7 (7)	93.82 ± 0.5 (7)
GFP Wee1	98.23 ± 2.3 (10)	99.1 ± 1.9 (10)	95.47 ± 2.6 (10)
FGF14 DMSO	100.8 ± 1.6 (14)	101.2 ± 1.4 (14)	101.3 ± 1.9 (14)
FGF14 Wee1	110.2 ± 3.5 (8) ^a^	120 ± 5.8 (8) ^b^	121.8 ± 5.5 (8) ^c^
FGF14^Y158A^ DMSO	105 ± 2.8 (11)	107.1 ± 4.3 (11)	106.4 ± 3.8 (11)
FGF14^Y158A^ Wee1	103.8 ± 3.4 (9)	108.9 ± 6.4 (9)	107.9 ± 6.5 (9)

^†^ Data are mean ± SEM; ns = nonsignificant; (*n*) = number of cells. ^a^ *p* = 0.021, One way ANOVA Tukey’s multiple comparisons test compared to FGF14 DMSO; ^b^ *p* < 0.0001, One way ANOVA Tukey’s multiple comparisons test compared to FGF14 DMSO; ^c^ *p* < 0.0001, One way ANOVA Tukey’s multiple comparisons test compared to FGF14 DMSO.

**Table 3 cells-10-03103-t003:** Effects of pharmacological inhibition of Wee1 kinase on Na_v_1.6-mediated currents ^†^.

Condition	Peak Density	Activation	K_act_	Steady-State Inactivation	K_inact_	Tau (τ)
	pA/pF	mV	mV	mV	mV	ms
GFP DMSO	−57.38 ± 6.3 (8)	−22.87 ± 1.69 (8)	4.79 ± 0.46 (8)	−60.4 ± 1.67 (8)	5.93 ± 0.48 (8)	1.03 ± 0.04 (8)
GFP Wee1	−62.99 ± 5.5 (8)	−24.38 ± 1.19 (8)	4.29 ± 0.43 (8)	−60.84 ± 1.67 (8)	6.15 ± 0.40 (8)	1.11 ± 0.04 (8)
FGF14 DMSO	−17.31 ± 2.5 (10) ^a^	−18.15 ± 1.03 (8) ^c^	5.05 ± 0.51 (8)	−55.77 ± 1.10 (8) ^d^	5.654 ± 0.57 (8)	1.64 ± 0.26 (8) ^e^
FGF14 Wee1	−20.57 ± 2.5 (10) ^ns,b^	−20.11 ± 1.25 (8)	5.89 ± 0.57 (8)	−57.12 ± 1.19 (8)	6.91 ± 0.80 (7)	1.53 ± 0.11 (9)

^†^ Data are mean ± SEM; ns = nonsignificant; (*n*) = number of cells. ^a^ *p* < 0.0001, One way ANOVA Tukey’s multiple comparisons test compared to GFP DMSO; ^b^ *p* < 0.0001, One way ANOVA Tukey’s multiple comparisons test compared to GFP Wee1; ^c^ *p* = 0.012, One way ANOVA Tukey’s multiple comparisons test compared to GFP DMSO; ^d^ *p* = 0.00090, One way ANOVA Tukey’s multiple comparisons test compared to GFP DMSO; ^e^ *p* = 0.0328, One way ANOVA Tukey’s multiple comparisons test compared to GFP DMSO.

**Table 4 cells-10-03103-t004:** Effects of Wee1 inhibitor II on entry of Na_v_1.6 channels into long-term inactivation ^†^.

Condition	LTI (% Maximal Na^+^ Current)		
	2nd Pulse	3rd Pulse	4th Pulse
GFP DMSO	94.11 ± 1.9 (8)	92.43 ± 1.3 (8)	92.18 ± 1.6 (8)
GFP Wee1	93.35 ± 2.1 (9)	91.02 ± 1.5 (9)	92.94 ± 1.5 (9)
FGF14 DMSO	101.0 ± 1.6 (8) ^a^	106.7 ± 4.0 (8) ^b^	104.5 ± 4.7 (8) ^c^
FGF14 Wee1	101.7 ± 2.2 (8)	104.6 ± 1.7 (8)	102.9 ± 1.9 (8)

^†^ Data are mean ± SEM; ns = nonsignificant; (*n*) = number of cells; ^a^ *p* = 0.0159, unpaired *t* tests compared to GFP DMSO; ^b^ *p* = 0.0012, One way ANOVA Tukey’s multiple comparisons test compared to GFP DMSO; ^c^ *p* = 0.0021, One way ANOVA Tukey’s multiple comparisons test compared to GFP DMSO.

## Data Availability

Data included in this study are available upon request from the corresponding author.

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
