# Peer review of "Pharmacological Inhibition of Wee1 Kinase Selectively Modulates the Voltage-Gated Na+ Channel 1.2 Macromolecular Complex"

_cells, 2021, doi:10.3390/cells10113103_

Round 1

Reviewer 1 Report

The authors here investigate how inhibiting the Wee1 kinase by Wee1 inhibitor II affects the Na channel macromolecular complexes. They found that Wee1 inhibitor II modulates the FGF14-Nav1.2 complex assembly but not the FGF14-Nav1.6. Also, the results show that Wee1 inhibitor II changes the electrophysical properties of Nav1.2 but not Nav1.6 in the presence of FGF14. These modulations are dependent on Y158 in FGF14. The data are convincing and the conclusions are appropriate. The manuscript is well-presented.

Author Response

Responses to the comments and suggestions from Reviewer 1:

Recommendation: Accept

1) The authors here investigate how inhibiting the Wee1 kinase by Wee1 inhibitor II affects the Na channel macromolecular complexes. They found that Wee1 inhibitor II modulates the FGF14-Nav1.2 complex assembly but not the FGF14-Nav1.6. Also, the results show that Wee1 inhibitor II changes the electrophysical properties of Nav1.2 but not Nav1.6 in the presence of FGF14. These modulations are dependent on Y158 in FGF14. The data are convincing and the conclusions are appropriate. The manuscript is well-presented.

Response: We thank this reviewer for the appreciation of our work. We are pleased to read this enthusiastic review of our work that endorses the manuscript for publication in the form in which it was originally submitted.

Reviewer 2 Report

This manuscript investigated how pharmacological inhibition of Wee1 selectively modulate Nav1.2 channel rather than Nav1.6 channel through electrophysiological recordings and overexpression HEK 293 system, which has the potential to unravel the underlying mechanism about how neuronal excitability is fine-tuned. The data were well-presented and easy to follow. I only have some minor comments.

  1. There are still a few typos spotted such as um (in Figure 1 legend, should be uM). The authors need to read carefully and correct them throughout the entire manuscript.
  2. Based on all data presented, it would be more appropriate to say FGF is functionally coupled to Na channels. The current terms such as FGF14:Nav1.2; FGF14:Nav1.6, Complex Assembly is misleading, as it gives the impression that FGF is biochemically coupled to Na channels, which is not supported by the presented experimental evidence. 
  3. Figure 1B is just a simple repeat of Figure 1A, both from the same data set; Figure 1D is just a simple repeat of Figure 1C (Figure 1F repeats Figure 1E). The authors should merge the same type of data together.
  4. From figure 2, 15 μM of Wee1 inhibitor II was used. Why the concentration of  15 μM was chosen needs to be clearly justified in the text.

Author Response

Responses to the comments and suggestions from Reviewer 2:

Recommendation: Minor revisions

1) This manuscript investigated how pharmacological inhibition of Wee1 selectively modulate Nav1.2 channel rather than Nav1.6 channel through electrophysiological recordings and overexpression HEK 293 system, which has the potential to unravel the underlying mechanism about how neuronal excitability is fine-tuned. The data were well-presented and easy to follow.

Response: We thank this reviewer for the appreciation of our work.

2) There are still a few typos spotted such as um (in Figure 1 legend, should be uM). The authors need to read carefully and correct them throughout the entire manuscript.

Response: We thank this reviewer for the close inspection of the text. This typo has been corrected. Additionally, we have closely inspected the manuscript for other aberrant spelling and corrected any issues.

3) Based on all data presented, it would be more appropriate to say FGF is functionally coupled to Na channels. The current terms such as FGF14:Nav1.2; FGF14:Nav1.6, Complex Assembly is misleading, as it gives the impression that FGF is biochemically coupled to Na channels, which is not supported by the presented experimental evidence. 

Response: We thank this reviewer for the feedback regarding the terminology employed in the manuscript. Referring to complexes involving iFGFs and Nav channels as FGF14:Nav1.2 and FGF14:Nav1.6 is widely employed in the literature (Ali et al., 2014, CNS Neurol Disord Drug Targets; Ali et al., 2016, JBC; Wadsworth et al., 2020, BBA; Wang et al., 2020, JMC; Dvorak et al., 2021, CTMC). In addition, crystal structures of iFGF:Nav complexes have been reported (Wang et al., 2012, Structure; Gardill et al., 2019, PNAS). Considering this broader context, using the terms FGF14:Nav1.2 and FGF14:Nav1.6, and referring to their complex assembly, is appropriate.

4) Figure 1B is just a simple repeat of Figure 1A, both from the same data set; Figure 1D is just a simple repeat of Figure 1C (Figure 1F repeats Figure 1E). The authors should merge the same type of data together.

Response: We thank this reviewer for the feedback regarding the representation of data in Figure 1. The figure has been updated accordingly. The bar graphs, which are essentially repeats of the dose-response curves, have been removed.

5) From figure 2, 15 μM of Wee1 inhibitor II was used. Why the concentration of  15 μM was chosen needs to be clearly justified in the text.

Response: We thank this reviewer for the inquiry regarding why a concentration of 15 µM was selected for electrophysiological studies. In the version of the manuscript originally submitted, this justification was provided in the third sentence of section 3.2. The pertinent text is included below for convenience:

 “Cells were then incubated for 30 min with either vehicle (0.1% DMSO) or 15 µM of Wee1 inhibitor II, a concentration selected on the basis of it being close to the IC50 value of the compound as determined in the LCA (Figure 1A).”

Reviewer 3 Report

The authors of the manuscript presented data indicating that Wee1kinase have exert selective regulation of the Na 1.2 channel complex buy not of the Na 1.6 channel complex. They suggest that this effect might be induced by phosphorylation of FGF14 at Y158 residue by Wee1 kinase.

This work is done very carefully and sounds very scientific. Although in vitro phosphorylation of FGF14 by the Wee1 kinase will be the final proof, it is already, in its present, form suitable for publication.

Author Response

Responses to the comments and suggestions from Reviewer 3:

Recommendation: Accept

1) The authors of the manuscript presented data indicating that Wee1kinase have exert selective regulation of the Na 1.2 channel complex buy not of the Na 1.6 channel complex. They suggest that this effect might be induced by phosphorylation of FGF14 at Y158 residue by Wee1 kinase. This work is done very carefully and sounds very scientific. Although in vitro phosphorylation of FGF14 by the Wee1 kinase will be the final proof, it is already, in its present, form suitable for publication.

 Response: We thank this reviewer for the appreciation of our work. We are pleased to read this enthusiastic review of our work that endorses the manuscript for publication in the form in which it was originally submitted.